# Jamming Transitions in Astrocytes and Glioblastoma Are Induced by Cell Density and Tension

**DOI:** 10.3390/cells12010029

**Published:** 2022-12-21

**Authors:** Urszula Hohmann, Julian Cardinal von Widdern, Chalid Ghadban, Maria Cristina Lo Giudice, Grégoire Lemahieu, Elisabetta Ada Cavalcanti-Adam, Faramarz Dehghani, Tim Hohmann

**Affiliations:** 1Department of Anatomy and Cell Biology, Martin Luther University Halle-Wittenberg, 06108 Halle (Saale), Germany; 2Department of Cellular Biophysics, Max Planck Institute for Medical Research, 69120 Heidelberg, Germany

**Keywords:** glioblastoma, migration, collective migration, jamming, unjamming, adhesion, tension, astrocytes

## Abstract

Collective behavior of cells emerges from coordination of cell–cell-interactions and is important to wound healing, embryonic and tumor development. Depending on cell density and cell–cell interactions, a transition from a migratory, fluid-like unjammed state to a more static and solid-like jammed state or vice versa can occur. Here, we analyze collective migration dynamics of astrocytes and glioblastoma cells using live cell imaging. Furthermore, atomic force microscopy, traction force microscopy and spheroid generation assays were used to study cell adhesion, traction and mechanics. Perturbations of traction and adhesion were induced via ROCK or myosin II inhibition. Whereas astrocytes resided within a non-migratory, jammed state, glioblastoma were migratory and unjammed. Furthermore, we demonstrated that a switch from an unjammed to a jammed state was induced upon alteration of the equilibrium between cell–cell-adhesion and tension from adhesion to tension dominated, via inhibition of ROCK or myosin II. Such behavior has implications for understanding the infiltration of the brain by glioblastoma cells and may help to identify new strategies to develop anti-migratory drugs and strategies for glioblastoma-treatment.

## 1. Introduction

Astrocytes are the most abundant cells in the central nervous system (CNS). Despite their numerous and region specific functions, astrocytes are also considered as support cells for neurons and are important for the maintenance of stable structures in the CNS, such as, e.g., the blood brain barrier [1,2,3]. To allow formation of stable structures under physiological conditions, astrocytes have to be mostly immobile. Nevertheless, under certain pathological conditions, e.g., during wound healing, astrocytes can become motile or as observed in astrocytoma or glioblastoma (GBM), transform into highly mobile and migratory tumor cells [4]. GBM belongs to one of the most deadly tumor entities, with a median survival time of approximately 14 months [5]. Besides its therapy resistance, GBM prognosis is poor due to its infiltrative nature, invading into the adjacent, healthy brain tissue [6,7,8]. Colonization of brain tissue happens in the form of single cells or in a collective manner, preferentially along pre-existing structures such as blood vessels or white matter tracts as branched networks or collective strands [9,10,11,12,13]. A complex self-interacting network of biochemical and biomechanical cues controls the dynamic of single cells and collective motility [14,15,16]. Regardless of the exact way of infiltration, a critical step for brain colonization is the attainment of a migratory phenotype of GBM cells that was not present in the cells of their origin. 

Earlier studies demonstrated that cells of epithelial origin can undergo such a transition and its inverse, similar to a (un-)jamming transition in soft matter, where cells attain an (im-)mobile phenotype [17,18,19,20,21,22]. Thereby, jammed matter has energy barriers between cells that are sufficiently high to hinder reorganization of the cell layer effectively, leading to fixed cell positions in relation to neighboring cells. The jamming transition is accompanied by a continuous decrease of cell speeds and an increase in the pack size of collectively moving cells [17,19,21]. One path to a jammed, mostly immobile state was governed by an increase in cell density [17,18,22,23,24,25]. In these studies, it was suggested that cell density regulates traction forces [24,25]. On the other hand different studies, both theoretical and experimental, found jamming to be additionally determined by the ratio of surface tension to cell–cell-adhesion [19,26,27,28]. Thereby, increasing tension led to jamming of the cell layer, while increasing adhesion had the opposite effect [19,26,27,28]. Still, the exact role of adhesion is controversially discussed as different studies found increased adhesion to cause jamming instead [20,22]. Data regarding the collective migration state of cells of glial origin is—to the authors’ knowledge—still lacking.

Nonetheless, the phenomenon of jamming might have a further functional role in glioma invasion as well, potentially being a mechanism for suppressing brain colonization. Supporting this idea, a study by Castro et al found that not only epithelial cells jam and mammary carcinoma cells remain unjammed, but also that the addition of epithelial cells causes a slowdown of carcinoma cells with increasing epithelial cell density [29]. A recent study performed in 3D collagen gels also demonstrated that increased adhesion and confinement of mammary carcinoma cells favors jamming [22]. In a retrospective study of mammary carcinoma, markers for unjamming were proposed as an additional, independent prognostic marker and associated with higher risk [30]. These observations raise the question if and how it is possible to induce a switch from a migratory, unjammed state to a more static, non-migratory and jammed state in glioma cells.

Here, we studied the collective behavior of primary astrocytes and two glioblastoma cell lines. As a result, we found astrocytes to reside in a jammed state, while glioblastoma cells remain unjammed, mostly independent of cell density. A switch from unjammed to jammed could be observed in one glioblastoma line after contractility and adhesion inhibition. 

## 2. Methods

### 2.1. Cell Culture

All animal experiments were conducted in agreement with directive 2010/63/EU of the European Parliament and European Council (22 September 2010) and tissue collection was approved by the local authorities of the state of Saxony-Anhalt (I11M18). 

For experiments, LN229 and U138 glioblastoma cells and primary astrocytes were used. LN229 cells were purchased from the American Type Culture Collection (ATCC, CRL-261, Manassas, VA, USA) and U138 cells were obtained from Cell Lines Service (Cell Lines Service, 300363, Eppelheim, Germany). LN229 cells were cultured using 89% (v/v) Roswell Park Memorial Institute medium (Lonza, Basel, Switzerland, BE12-115F), supplemented with 10% (v/v) fetal bovine serum (FBS, Gibco, Carlsbad, CA, USA, 10500-064) and 1% (v/v) penicillin/streptomycin (P/S, Gibco, Carlsbad, CA, USA, 15140-122). U138 and astrocytes were cultured in 89% (v/v) Dulbecco’s Modified Eagle Medium (Invitrogen, Waltham, MA, USA, 41965-062), and supplemented with 10% (v/v) FBS and 1% (v/v) P/S. 

Primary astrocytes were isolated from 1–2 days old Black6/J mice, as described previously [31]. Briefly, mice were decapitated, brains collected and meninges removed. Afterwards, brains were dissociated in Ca^2+^/Mg^2+^ free HBSS (Gibco, Carlsbad, CA, USA, 12082739), containing trypsin (Gibco, Carlsbad, CA, USA, 12579069) and DNAse (Worthington Biochemical, Bedford, MA, USA, LS006342). The cell suspension was then transferred to cell culture flasks. For all experiments, astrocytes of passage number one to three were used.

For experiments analyzing collective migration, 300,000–800,000 cells were seeded into one 12-well. In case of astrocytes, an artificial wound was generated scratching the monolayer as reported [32]. Blebbistatin (Tocris, Bristol, UK, 1760) and Y-27632 (Tocris, Bristol, UK, 1254) were used to inhibit myosin II and ROCK, respectively. Substances were applied 1 h prior to the measurements for spheroid aggregation and measurements of collective cell migration or 24 h before measurements for all other types of experiments, to assess equilibrium conditions. Before starting the experiments, culture medium was replaced with fresh medium, containing the respective drugs. All experiments were performed in the presence of the respective treatments. 

### 2.2. Measuring Properties of Collective Cell Migration

Twenty-four hours after cell seeding, cells were transferred to an inverted microscope (DMi 8, Leica, Wetzlar, Germany) equipped with temperature (37 °C) and CO_2_ (5% (v/v)) control. Images were taken every 3 min for 20 h or 60 h and denoised using block matching 3D transform [33]. To analyze the velocity field of a cell layer, particle image velocimetry [17,19,20] was used with a cross-correlation window size of 16 × 16 pixels (pixel size: 0.48 µm). Resulting trajectories closely approximate cellular motions [19,20,34,35]. 

Furthermore, the self-overlap function Q(Δt) [36] was calculated to further quantify cellular movement, as described previously [34]:$$Q\left(\Delta t\right)=\frac{1}{N}\overset{N}{\underset{i=1}{\sum }}w_{i}\;\;\;with\;w=\{\begin{array}{c} 1;if\;\Delta r>0.2d\\ 0;else\;\;\;\;\;\;\;\;\;\;\;\;\;\;\;\;\;\;\; \end{array}$$

Here, *N* depicts the cell number and Δ*r* is the distance of a virtual particle to its initial position and *d* the cell diameter. For reasons of comparability, d = 80 px (≈38.4 µm) for all cell types was chosen, corresponding to the cell diameter of astrocytes, the largest of the three used cell types. *Q* gives the relative number of cells that moved away more than 20% of its cell size from their initial position. For quantification of cooperativity, the 4-point susceptibility *χ* was calculated:$$\chi =N\left[\langle Q{\left(\Delta t\right)}^{2}\rangle -\langle Q{\left(\Delta t\right)\rangle }^{2}\right]$$

The peak height of *χ* is proportional to the number of collectively moving cells in a dense layer, and the peak position corresponds to the pack lifetime [23,36]. For assessing the shape of cells, cells from the time lapse images were segmented manually every 4 h to calculate the shape index *p* proposed by Bi et al [26], using the cell’s projected area *A* and its perimeter *P*:$$p=\frac{P}{\sqrt{A}}$$

From the obtained cell areas, cell density was estimated. Furthermore, to assess caging for each non-boundary virtual particle, it was calculated how many of its eight nearest neighbors at the beginning and end of each measurement were unchanged.

Furthermore, the spatial velocity–velocity auto-correlation length was calculated as the characteristic exponential decay length of the auto-correlation function *C* of the velocity $\vec{v}$, defined at time *t*, and position $\vec{x}$ as:$$C{\left(\Delta \vec{x}\right)}_{t}=\frac{\langle \vec{v}\left(\vec{x}\right)*\vec{v}\left(\vec{x}+\Delta \vec{x}\right)\rangle }{\langle \vec{v}\left(\vec{x}\right)*\vec{v}\left(\vec{x}\right)\rangle }$$

### 2.3. Measuring Properties of Single Cell Migration

For time-lapse microscopy, 1000 cells were seeded in a 12-well plate (Greiner, Kremsmünster, Austria) 24 h prior to the start of experiments. Isolated single cells were imaged every 15 min with a microscope (Leica DMi8, Leica, Wetzlar, Germany) equipped with CO_2_ (5% (v/v)) and temperature (37 °C) regulation. From these measurements, the average cell speed and directionality were calculated. The directionality was defined as the quotient of the total distance travelled by a cell and the sum of incremental distances the same cell moved between successive frames.

### 2.4. Determination of Cortex Tension by Atomic Force Microscopy

For measuring the surface tension, sparsely seeded cells were allowed to adhere to a petri dish for 15 min before the experiment. Single cells were measured with a tip-less cantilever (Arrow-TL2, Nanoworld, Neuchatel, Switzerland), applying a force of 1 nN for determining the cortex tension, as described by Cartagena-Rivera et al. [37,38]:T=kπ(1Zd−1)

Here *T* is the cortex tension, *k* the elastic constant of the cantilever, *Z* the piezo extension and *d* the deflection of the cantilever. The cortical tension is considered the upper limit of a force generated by the actin cortex to limit cell deformations. The described experimental setting and all other types of atomic force microscopy measurements are summarized in Appendix A. 

### 2.5. Determination of Elastic Modulus by Atomic Force Microscopy

Additionally, the Young’s modulus of a dense cell layer was measured over a 140 × 140 µm area with a step-size of 7 µm using a CP-qp-SCONT-BSG-A (NanoAndMore, Wetzlar, Germany) cantilever with a 5 µm bead attached to it. The Young’s modulus was calculated using the Hertz model:$$F=\frac{4}{3}\frac{E}{1-\mu^{2}}\sqrt{R\delta_{0}^{3}}$$

With the applied force *F* (here: 1 nN), the Young’s Modulus is *E*, Poissons ratio *µ*, bead diameter *R* and indentation *δ*_0_.

### 2.6. Determination of Adhesion Energies by Atomic Force Microscopy

Using AFM, the cell–cell adhesion energy between a single cell and a dense cell-layer of the same cell type and treatment was measured. A tip-less cantilever (Arrow-TL2, Nanoworld, Neuchatel, Switzerland) was coated with poly-L-lysine overnight. For cell attachment, the cantilever was pressed onto a freshly suspended, non-adherent cell with 1 nN for 30 s. The cell was then allowed to firmly attach to the cantilever for another 30 min. When firmly attached, the cells elastic modulus was determined and cell–cell adhesion was measured by pressing the cantilever with the cell for 60 s on multiple spots of a dense cell-layer with a force of 0.5 nN. From the obtained retract curves the maximal force of adhesion was calculated. To assure mechanical stability of the attached cell, the Young`s modulus of the attached cell was measured again at the end of all measurements. The measurement set was discarded in cases of deviations of larger than 20% between the initial and final elastic modulus.

### 2.7. 3D Tumor Aggregate Formation Assay

For cultivation of 3D tumor aggregates, the liquid-overlay method was used. Therefore, 50,000 cells were plated in 96-wells coated with 4% (w/v) agarose, to generate a non-adhesive surface. The plate was shortly centrifuged (50 g, 4 min) and cells were allowed to aggregate for 2 h before starting the imaging process. The delay was necessary to determine the final position of an emerging aggregate. Imaging was performed for up to 68 h, and images were taken every 15 min. As read-outs, the aggregate size and its brightness relative to the background were determined. 

For image analysis, we used a custom written MatLab (The MathWorks, Natick, MA, USA) script determining the edge of the 3D aggregate using the Chan-Vese image segmentation model [39], tracking each 3D aggregate over time [40]. 

Together with the AFM measurements, the aggregation assay allows the estimation of the cell–cell adhesion energy *W* [40,41], using the following formula:$$W=\frac{2Ed}{\sqrt{3}\pi \left(1-\nu^{2}\right)}{\left(1-{\left(\frac{A_{\;\infty }\;\left(1-I_{\infty }\right)}{A_{\;0}\;\left(1-I_{0}\right)}\right)}^{1/3}\right)}^{3/2}$$

Here, *E* denotes the Young’s modulus, *d* the cell diameter, *ν* the Poisson ratio, *A*_0_ and *A_∞_* the initial and equilibrium projected spheroid area and *I*_0_ and *I_∞_* the initial and equilibrium relative brightness. Similarly, as done by Frasca et al. [41], adhesion energy was extracted from the adhesion over time plot after reaching a stable plateau, as the average of the last 50 time points. Reaching of equilibrium was assessed by inspection of the projected spheroid area. The derivation of the model is described in more detail below.

For estimation of the cell–cell adhesion using the aggregation assay, we used a modified model introduced by Frasca et al. [40,41]. There, it was concluded that the adhesion energy *W* for establishing cell–cell contact is given as:$$W=\frac{2Ed}{\sqrt{3}\pi \left(1-\nu^{2}\right)}{\left(1-{\left(\frac{p_{0}}{p_{\infty }}\right)}^{1/3}\right)}^{3/2}$$

Here, *E* is the Young’s modulus of a single cell, *d* the initial cell–cell distance, *ν* the Poisson ratio, *p*_0_ and *p_∞_* the initial and equilibrium compacity. The compacity describes the compactness of the 3D aggregate and is given as:$$p=N*\;\frac{V_{Cell}}{V_{Sphaeroid}}$$
with *N* as the number of cells in the aggregate and *V* as the volume of a cell or 3D aggregate, respectively. 

The Young’s modulus *E* was measured with the AFM, along with the average cell diameter, being a very good estimate of the initial cell–cell distance *d*. This assumption is valid, as the diameter of the cells was measured for AFM experiments 15 min after seeding. The 3D aggregate volume is given by *V = A*h*, with the 3D aggregate projected area *A* and its average height *h*. The area *A* was directly extracted during the aggregation experiments, while the height *h* can only be indirectly assessed, using: *h* ~ 1-*I*, with the measured relative intensity *I*. 

Consequently, the energy of adhesion is given as:$$W=\frac{2Ed}{\sqrt{3}\pi \left(1-\nu^{2}\right)}{\left(1-{\left(\frac{A_{\;\infty }\;\left(1-I_{\infty }\right)}{A_{\;0}\;\left(1-I_{0}\right)}\right)}^{1/3}\right)}^{3/2}$$

This equation assumes that N_0_*V_cell 0_ ≈ N_∞_*V_cell ∞_. 

### 2.8. Fluorescence Staining

For analysis of actin structures, cells were labeled with phalloidin and 4′,6-Diamin-2-phenylindol (DAPI) to stain nuclei. ß-Catenin and N-Cadherin were used for visualization of cell contacts. An amount of 30,000 cells were placed on glass coverslips to create a dense cell layer and incubated for 24 h prior to treatment. Twenty-four hours after treatment, cells were fixed with 4% paraformaldehyde for 10 min. For fluorescence labelling, normal goat serum was applied for 30 min, before incubation with primary antibodies for 16 h (for human tumor cells: N-Cadherin, 1:500, MA5-29138, Invitrogen, Waltham, MA, USA or for murine astrocytes and tumor cells: ß-Catenin, 1:200, 71–2700, Thermo Fisher, Waltham, MA, USA). On the next day and after washing with PBS, the secondary antibody (goat anti-rabbit-Alexa 568, 1:200, A11011, Invitrogen, Waltham, MA, USA) was applied for one hour. Next, for actin labelling, a phalloidin-488 staining was used. Cells were washed twice for 10 min in PBS, then incubated with 0.1% PBS/Triton for 5 min and blocked with 1% bovine serum albumin. Phalloidin-488 (2.5 µL/100 µL BSA solution, Thermo Fisher Scientific, Waltham, MA, USA, A12379) was applied for 20 min. For the visualization of nuclei, DAPI (1:10,000, Sigma Aldrich, St. Louis, MO, USA D9542) was used. The stained cells were washed with PBS, distilled water and covered with DAKO mounting medium (DAKO, Santa Clara, CA, USA).

Fluorescence images were acquired with a 63× objective using a confocal laser scanning microscope (Leica, DMi8, Wetzlar, Germany). For detection of DAPI, phalloidin, β-catenin and N-cadherin, the following excitation wavelengths were used: 405 nm, 488 nm, and 568 nm, respectively. Emission was detected in the range of Δλ = 420–500 nm (DAPI), Δλ = 510–550 nm (Phalloidin) and Δλ = 580–680 nm (β-catenin, N-cadherin).

For the analysis of cell–cell-contacts in β-catenin, staining contact regions with an approximately straight shape were chosen. The cell–cell contact was manually marked and the mean intensity inside a rectangle along the direction parallel to the cell–cell contact was measured. The background was normalized and Gaussian fitting performed. As the strength of a cell–cell contact is supposed to be proportional to its intensity and thickness, the adhesive index was calculated as the sum of the normalized intensity around the peak in a distance ± two standard deviations away from the peak center.

### 2.9. Estimation of Stress Fiber Tension

For an estimate of the tension created by stress fibers, it was used that the work *W* needed to increase the cell area *A* is given as:dW=T∗dA
where *T* denotes the tension. In equilibrium, if the cell–surface area does not increase further, the work conducted for cell spreading is equivalent to the energy stored by the material of the cell and its contractile elements (stress fibers and cortex). The contribution of the stress fibers can be estimated using the average force *F* created by stress fibers, the length *L* over which the force can be transmitted and the number of stress fibers *N*:W=F∗L∗N

Thus, for a cell of contact area *A* the tension can be estimated as follows:$$T=\frac{F*L*N}{A}$$

This approach assumes that the generated force per stress fiber is constant along the whole distance if the fiber retracts, as well as a roughly uniform distribution of stress fibers. Furthermore, it assumes all stress fibers to behave equally. Thus, this approach likely gives an overestimation of the generated tension, as the force generated by stress fibers decreases during contraction. Consequently, the values obtained should be considered upper bounds for forces generated by stress fibers to resist shape changes.

From the literature, the force of contractile fibers in reconstructed actin–myosin bundles was estimated to be 0.5–2 nN [42,43], and measurements in U2OS cells and simulations found values around 2–4 nN [44,45], while retraction length of stress fibers upon laser ablation in glioma cells and others was ~2 µm [46,47], corresponding to half the upper limit a stress fiber can transmit a force. Here, we assume a force of 3 nN per fiber and a contraction length of 4 µm. The numbers of stress fibers per cell and cell–cell contact area were extracted manually from laser scanning microscopy images of dense cell layers.

### 2.10. Traction Force Microscopy

Double layer 35 kPa hydrogels optimized for TFM were prepared as follows. 24 mm glass coverslips were silanized by 15 min incubation in a solution with a ratio of (14:1:1) of ethanol 96%, acetic acid and 3- (Trimethoxysilyl)propyl methacrylate (Sigma Aldrich, St. Louis, MO, USA, #440159). Afterwards, the coverslips were rinsed with ethanol and dried. A 35 kPa Acrylamide/Bisacrylamide stock solution was prepared by mixing 2.5 mL of 40% Acrylamide (Bio-Rad, Hercules, CA, USA, #161-0140), 1.5 mL of 2% Bisacrylamide (Bio-Rad, Hercules, CA, USA, #161-0142) and 6 mL of MilliQ water. An amount of 99.3 µL Acrylamide/Bisacrylamide stock solution was mixed with 0.5 µL of 10% Ammonium persulfate (APS, Sigma-Aldrich, St. Louis, MO, USA, #A3678) and 0.2 µL N,N,N′,N′-tetramethylethylenediamine (TEMED, Bio-Rad, Hercules, CA, USA, #161-0800). An amount of 5 µL of the mix was put on top of the silanized 24 mm coverslip and a 15 mm glass coverslip was put on top of the drop. After 30 min, the top coverslip was removed. An amount of 94.3 µL of the Acrylamide/Bisacrylamide stock solution was then mixed with 5 µL fluorescent beads (Invitrogen, Waltham, MA, USA, F8807, 0.2 µm, dark red fluorescent), 0.5 µL of 10% APS and 0.2 µL TEMED. An amount of 5 µL of this solution was put on top of the first hydrogel and a 15 mm round glass coverslip was placed on top. After 30 min, the second coverslip was removed, and the TFM hydrogel surface was activated by incubation with 250 µL of 1 mg/mL N-sulfosuccinimidyl-6-(4′-azido-2′-nitrophenylamino) hexanoate (SulfoSANPAH, Sigma-Aldrich, St. Louis, MO, USA, #803332) in MilliQ water, followed by 6 min of UV exposure. The hydrogels were then incubated with 100 µg/mL laminin for 1 h, placed in 6-well plate holders and kept in PBS containing 1% Penicillin-Streptomycin (Gibco # 15140122) overnight. The next day LN229 cells were seeded on the hydrogels at a density of 10^6^ cells per well and let to adhere for 24 h. 

Samples were imaged using a Nikon Eclipse TI2 microscope equipped with a 40× Plan Apo air objective. To investigate the influence of blebbistatin and Y-27632 treatment on LN229 traction forces, samples were pre-treated for 1 h prior to imaging with 5 µM blebbistatin or 20 µM Y-27632. A total of 5–10 areas were selected for each sample and Z-stacks were acquired before and after the addition of 20 µL 10% sodium dodecyl sulfate. Bead displacements were detected using the cross-correlation PIV plugin in ImageJ after stacking and alignment using the SIFT alignment algorithm [48]. Heatmaps of bead displacements were coded in MATLAB (version R2020a). Please denote that only bead displacements and not tractions were measured, albeit larger displacements imply larger traction forces.

### 2.11. Statistics

Statistics were performed using the two-tailed ANOVA with the Tukey post-hoc test or the two-sided sign test to evaluate if the median of the shape factor is above or below a critical value. Significance was defined for *p* < 0.05. All error-bars and shaded areas depict the standard error of the mean. Experiments were repeated at least three independent times.

For live-cell imaging experiments assessing collective motion, 5–8 fields of views were measured per experiment and condition. For primary astrocytes, each individual live-cell experiment was performed with astrocytes from different animals to avoid passaging effects. 

For AFM measurements of dense layers, three or four independent experiments were performed capturing one measurement field with 400 individual measurement curves per experiment. Data points were excluded, when the approach curve contained kinks or shape variations that point towards cantilever or cell slippage.

Confidence intervals for fit parameters were obtained using the MatLab function confint.

## 3. Results

### 3.1. Astrocytes Are Non-Migratory and Jammed but Not Glioblastoma

We first measured the collective motion of primary astrocytes and the human GBM cell lines U138 and LN229. All cell types showed different behavior (Figure 1), with astrocytes being less motile and mostly stationary, while the GBM cell lines showed reorganization and high motility (Figure 1, Appendix A). This observation agreed with the quantitative analysis of sheet migration. Astrocytes moved slow (5–6 µm/h), while GBM cells moved faster (8–17 µm/h; Figure 2a). In agreement, the order parameter of astrocytes showed high overlap during the measurement window of 20 h, with ≈70% of astrocytes located closer than 0.2 cell diameters to their initial position (Appendix A). In contrast, only ≈20% or less than 10% of the GBM cells were close to their initial position after 20 h, indicating a more static behavior of astrocytes compared to the GBM cells (Appendix A). From the order parameter, the 4-point susceptibility was calculated (Figure 2b). Its peak height is proportional to the number of cells moving coordinated and collectively, referred to as “pack” and the peak time corresponds to the time a pack of fast moving cells moves together. After several pack life times, the layer underwent significant reorganization. Analyzing the 4-point susceptibility, astrocytes and U138 cells were found to move in packs of 2–3 cells, while LN229 cells moved in packs of around 10 cells. As expected from the previous results, the pack lifetime averaged over each field of view was largest for astrocytes (11 h), and lower for the GBM cell lines (LN229: 5.5 h and U138: 1.5 h, Figure 2b). Tracking virtual particles and analyzing their relations in terms of changes in nearest neighbors, GBM cells had significantly different neighborhoods at the beginning and end of the measurement (less than 30% or 5% conserved neighborhood) compared to primary astrocytes (≈70% conserved neighborhood, Figure 2c). GBM cell monolayers showed, therefore, significant reorganization during the measurement time, while astrocytes remained relatively stable in their organization. Altogether, the obtained data indicated that astrocytes were in a jammed state, while the analyzed GBM cells were not. In previous studies based on the vertex model, a critical shape factor of 3.81 for the onset of jamming was derived [19,26,27]. Thus, the shape factor was calculated and found to be above the critical value for both GBM cell lines and below for astrocytes (Figure 2d), indicating that the path to jamming potentially follows the vertex model. Additionally, the shape factor stayed nearly constant during the whole measurement time (Appendix A, control conditions).

### 3.2. Cell Density Affects Jamming in a Cell Line Dependent Manner

Next, cell density was investigated as a possible path to (un-)jamming in the examined cell types, as it was proposed to be a major driver of jamming [17,49]. After visual inspection of the live cell videos, no significant proliferation was observed in astrocytes during the 20 h measurement time. Thus, different amounts of astrocytes were seeded to alter cell density. As an extreme case, a scratch assay was performed to generate a local astrocyte cell density of zero at the scratched region. GBM cells displayed a high amount of proliferation events during the measurement, increasing cell density up to a factor of three (Figure 3e–f). Thus, it was sufficient to measure GBM cell density at different time points. 

For astrocytes, no significant changes were found regarding cell speed, pack-size of collective motion (2–3 cells) and shape when altering cell density, except for the extreme case of the scratch (Figure 3a–d). Only an increase in pack lifetime from 11 h up to 15 h was found, indicating even slower reorganization for higher cell densities. Furthermore, the shape factor remained constant over time and the neighborhood of astrocytes was almost constant at the beginning and end of the measurement, independent of cell density (Appendix A). Induction of the scratch led to increased cellular motion (7–8 µm/h), with pack-sizes of up to 9 cells and significantly elongated cell shapes of astrocytes, implying unjamming (Figure 3a–c). Interestingly, if the scratch was closed the velocity field at the contact zone of the fronts was oriented roughly perpendicular towards the initial flow or there was only very little residual motion and the fronts did not mix (Figure 4a,b), implying jamming after wound closure. This assumption was strengthened by the observation that astrocytes slightly away from the meeting points of cell fronts (200–400 µm) regained a shape factor of 3.81 and even the more motile cells in the center showed less elongated shapes with a shape factor of 3.96 (Figure 4c).

As GBM cells divided frequently during the measurement, the 20 h measurements were subdivided into 6 h long time-windows to analyze the cell-density dependence of the 4-point susceptibility (Appendix A). Thereby, time averaged measured pack lifetimes matched the lifetimes of the 20 h time window only if the maximal lag time was at least 2.5 to 3 times larger than the actual pack lifetime. Notably, this was not the case for LN229 cells, using the 6 h long time-window. Longer lag times on the other hand would lead to a too large averaging effect over time and thus over cell density. See Appendix A for more information. Thus, only the ratio between pack life times at t = 0–6 h and t = 15–21 h is given here. Thereby, a monotonous increase over time of the pack lifetime up to 45% and up to 29% was observed for LN229 and U138 cells, respectively. Furthermore, a decrease of the peak height was observed for LN229 cells, while it stayed almost constant for U138 cells (Appendix A). The decrease of the peak height for LN229 cells might be explained by the increasing pack lifetime successively approaching the maximal lag time of 6 h, in an identical fashion as explained in a previous study [36] resulting in a decrease of the peak height. Thus, this was likely an effect of the chosen time frame. 

Furthermore, the speed of LN229 cells scaled with cell density as $v\;\sim \;1/\sqrt{\rho }$ and for U138, it was independent of density (Figure 3e–f). Such scaling behavior as for LN229 cells was predicted previously, when cell speed was determined by an equilibrium of expansive and contractile forces [24,50]. Notably, if the cell density was increased further via experimental times up to 60 h, the described behavior for U138 cells did not change (Appendix A). In contrast, the characteristics for LN229 cells approached those of astrocytes, with reorganization times of 13 h and a similarly conserved neighborhood as astrocytes with up to 80% conservation in a 20 h time window (Appendix A). Due to the very high cell densities, single cells could no longer be distinguished and cell density analysis was not performed. Taken together, cell density seemed to be a possible inducer of jamming in GBM cells, but in a cell line dependent manner. Next, we investigated additional cell-density independent mechanisms involved in jamming of GBM cells. Therefore, a disturbance of the force balance was induced. 

### 3.3. ROCK and Myosin II Inhibition Causes Jamming in Glioblastoma Cells

For inhibition of force generation and thus adhesion and tension [51,52], the myosin II inhibitor blebbistatin (5–20 µM) and ROCK inhibitor Y-27632 (5–40 µM) were used in different concentrations. Application of inhibitors reduced the collective migration speed by up to 50% for astrocytes and increased the pack lifetime from 12 h up to 15 h and pack size from ≈2 cells up to ≈3 cells (Figure 5a,b). Furthermore, layer reorganization stayed low with about 60–70% of the cells having a conserved neighborhood during the measurement time (Figure 5c). Consistently, all shape factors remained below the critical value for unjamming for the whole measurement time (Figure 5d, Appendix A). 

Application of the inhibitors to the monolayer of LN229 cells led to a drop of the speed by up to 40% and an increase of the pack lifetime up to 8 h (for both: except for 10 µM blebbistatin), accompanied by an increase of the 4-point peak susceptibility, resulting in an increase of the collectively moving cells from 10 to 18 (Figure 5e,f). Additionally, treated LN229 cells showed less layer reorganization, with up to 60% of cells not making new neighbors (Figure 5g, Appendix A), similar to the values observed for astrocytes. Furthermore, the shape factor dropped below the critical value for LN229 cells treated with 5 µM and 10 µM Y-27632, while it could not be statistically distinguished from the critical value for 20 µM Y-27632 and 5 µM blebbistatin (Figure 5h). Taken together, these observations indicate the onset of jamming. After treatment with 10 µM blebbistatin, LN229 cells lost most cell–cell-contacts, collapsed to a roundish shape with long thin processes, leaving significant space between individual cells, while moving more independently and crawled above each other. Thus, the assumptions for the analysis of collective motion and jamming were no longer met. When analyzing the effect of both inhibitors on single cell motility of LN229 cells, blebbistatin reduced single cell speed, while Y-27632 increased motility. Both inhibitors reduced the directional persistence of motion (Appendix A). 

For U138 cells, inhibition of force generation resulted in reduced collective migration speeds (up to 50%, Figure 5i), but there was still strong reorganization of the cell layer (Figure 5k, Appendix A) and collective motion (pack size: 2–4 cells; Figure 5j). Even though an increase in pack lifetime was observed, it reached at most 2.5 h (Figure 5j). Furthermore, the shape factor of U138 cells was reduced upon treatment, but remained well above the critical value (Figure 5l). The effects of the inhibitors on the single cell level were different, resulting in a marked increase in single cell speed (Appendix A). 

Consequently, disturbing the equilibrium of expansive and contractile forces induced jamming in LN229 cells, but not in U138 cells. As the effects of both inhibitors on single cell motility was different compared to the effect on the cell-collective, it was next tested how cell–cell-interactions were altered. 

### 3.4. (Un-)Jamming Is Determined by the Ratio of Adhesion and Tension

To identify the biomechanical origins of the jamming transition, multiple models were suggested. One model associated the transition from a fluid-like to a solid-like behavior by adhesion maturation of cell–surface- and cell–cell adhesions and thus by rising friction, assessed via velocity auto-correlation and layer speed [20]. Here, comparably low auto-correlation length was found in the order of one cell size (10–20 µm) that remained approximately constant over time, even for the LN229 cells (Appendix A). Qualitatively, the observed behavior did not change after application of ROCK or myosin II inhibitors for all cell lines and only slight numerical changes were observed (not shown). As the changes in auto-correlation length were very small compared to the intra-experimental variations and the values presented in Garcia et al. (±100 µm; here: ±10 µm), the proposed model may not be sufficient to explain changes in instantaneous velocity and layer reorganization. 

Another model explained the onset of (un-)jamming with alterations in the ratio of force generation to cell–cell-adhesion [26,27]. Thus, the force generated by stress fibers was estimated and cortical tension, elastic modulus and cell–cell-adhesion of the used GBM cell lines and astrocytes were measured. 

First, the elastic modulus of LN229 cells was measured after treatment with 20 µM Y-27632 or 5 µM blebbistatin and the median Young`s modulus decreased from 812 Pa to 288 Pa or 581 Pa, respectively (Figure 6a,b). Similarly, for U138 cells, median values declined from 526 Pa to 324 Pa (40 µM Y-27632) or 390 Pa (10 µM blebbistatin). Astrocytes were stiffest with a median modulus of 900 Pa (Figure 6b). When assessing cortical tension, a drop was observed after application of the inhibitors for both GBM cell lines, but changes were comparably small. In LN229 cells, median tension was reduced from 328 pN/µm to 237 pN/µm or 335 pN/µm and for U138, from 120 pN/µm to 99 pN/µm and 85 pN/µm, respectively. Again, tension of astrocytes was highest with a median of 764 pN/µm (Figure 6c). The reduction in cortex tension for the GBM cells was also in agreement with the observation that both inhibitors led to a reduction in the number and thickness of stress fibers (Figure 6d, Appendix A). Using a strongly simplified model for the estimation of the tension generated by stress fibers, a similar decrease was found. Median tension values were 727 pN/µm, 318 pN/µm and 401 pN/µm for LN229 and 575 pN/µm, 213 pN/µm and 280 pN/µm for U138 cells under the given conditions (Figure 6e). Values for astrocytes were highest with 1120 pN/µm. To verify the results from the previous estimates, traction force microscopy (TFM) measurements were performed. For LN229 cells, the bead displacements were lowered to 41% or 71% of the control values, implying significantly lower tractions (Figure 6f,g). U138 cells failed to adhere to laminin-coated polyacrylamide surfaces and therefore, no TFM measurements were performed. Afterwards, cell–cell-adhesion was assessed.

Both GBM cell lines functionally express the cell–cell-adhesion molecule N-cadherin at their cell periphery (Appendix A). As other cadherins might also be involved in the formation of cell–cell-junctions, the adaptor protein β-catenin was labelled to estimate relative cell–cell-contact strength. After the application of both inhibitors, a drop in the fluorescence intensity of the β-catenin staining became visible in both GBM cell lines and cell–cell–surfaces displayed no longer a clear continuous but a discontinuous, dot-like labeling (Figure 7a, Appendix A). The quantitative analysis confirmed this observation and the relative cell–cell-contact fluorescence intensity was reduced from 42.9 to 11.3 or 15.2 for LN229 and from 39.4 to 15.6 or 17.4 for U138 after blebbistatin or Y-27632 treatment, respectively (Figure 7b). Another type of adhesion experiment was performed using the AFM, pressing a single cell onto a dense cell layer for 1 min. Measured astrocytes had a median maximal adhesion force of 0.84 nN (Figure 7c). The adhesion force of LN229 cells decreased from 1.05 nN to 0.83 nN or 0.95 nN and of U138 cells from 1.59 nN to 1.3 nN or 0.91 nN after the application of inhibitors (Figure 7c). Lastly, the cell–cell-adhesion energy was calculated from a combination of the measured Young’s modulus (Figure 6b) and spheroid aggregation experiments (Figure 7e, Appendix A). Thereby, we obtained median values of 1299 µJ/m^2^ for astrocytes and 1698 µJ/m^2^, 281 µJ/m^2^ and 567 µJ/m^2^ for LN229 and 1732 µJ/m^2^, 769 µJ/m^2^ and 971 µJ/m^2^ for U138 cells (Figure 7d) after application of the ROCK or myosin II inhibitors. Notably, the adhesion ratios for LN229 and U138 after treatment to the respective controls were very similar for the spheroid experiments and the analysis of the fluorescence intensities (Appendix A). 

Thereafter, changes in adhesion were compared with those in tension for LN229 cells under equilibrium conditions (Table 1). The treatment with Y-27632 impacted adhesion stronger than tension. Whereas the adhesion was reduced to 16-26% of control values, tension was reduced to 38%. A similar observation was found for treatment with Blebbistatin, reducing adhesion to 33–35% and tension to 71% of control values. For U138, adhesion and tension were affected to a similar extent. Notably, tension estimates for U138 could only be performed indirectly.

Such shifts in ratios in favor of tension imply, together with the other presented data, a jamming transition or a layer close to the transition point for LN229. Similar changes were not observed for U138. For all cases, these predictions matched the observations made during live cell imaging.

## 4. Discussion

In this study, the collective behavior of primary astrocytes and GBM cells were analyzed. Astrocytes were found to reside in a jammed state allowing only little reorganization, while GBM cells were unjammed and showed marked reorganization. Interestingly, modulating cell density had a cell type and cell line dependent effect on the behavior of astrocytes and GBM cells. Furthermore, a perturbation of the equilibrium of force generation and adhesion induced jamming in LN229 cells. 

From a physiological point of view, the observed behavior appears plausible, as astrocytes naturally form and support structures that need to be temporally stable, such as synapses or the blood brain barrier [1]. On the other hand, GBM cells readily colonize the whole brain, demanding vivid reorganization of the tumor tissue. Nevertheless, what physical properties of the cells may cause the switch between a static, solid-like behavior as observed for astrocytes and a migratory, fluid-like behavior (GBM cells) have been investigated only sparsely. In cells of mostly epithelial origin, an increasing cell density was found to be a major inducer of jamming [17,18,22,23,24,25]. In our study, cell density appeared to be one control parameter for the induction of migratory arrest in one of the used GBM cell lines, while the other remained unaffected. This observation is in agreement with the findings of Castro et al showing the speed of mammary carcinoma cells in a dense layer to be mostly independent of the cell density, as observed here for U138 [29]. Nonetheless, we found a density dependent decrease in cell speed for the LN229 cells, scaling in a similar way as reported before, associated with the balance of traction forces and cell–surface frictional forces [24,50]. However, the effect of increased cell density was sufficient to cause jamming only at very high cell densities where no precise cell number estimates could be made. Therefore, in LN229 the scaling behavior observed for lower densities cannot be verified at higher densities associated with migratory arrest. Another aspect of possible importance is the maturation of adhesions over time, being related to cellular slow down and jamming [20]. This phenomenon appears less important for U138 cells, as all dynamic measurements (collective migration, spheroid aggregation) showed no large dynamics over time and spheroids gained their equilibrium size after less than 20 h with no significant changes afterwards. Similarly, if adhesion maturation strongly affects LN229, it is expected that spheroid aggregation does not reach a stable equilibrium after ≈24 h. Furthermore, collective migration experiments started 24 h after cell seeding; thus, it is likely that adhesions had already matured. Despite adhesion maturation, different ratios in cell–cell to cell-substrate adhesion might have played a role, especially for LN229 cells. Analyzing single cell motility, being only influenced by cell–substrate friction, velocities under treatment were up to two times higher. With growing cell density—given volume conservation of cells—the ratio of cell–cell contact area to cell–surface contact area increased and thus cell–cell friction became increasingly important [20,24]. While this approach might explain effects in LN229 cells, it seems insufficient to explain cell density independence in U138. Consequently, jamming in cells of glial origin might not only be governed by cell density, but disruptions in the equilibrium of frictional (adhesive) and traction forces are considered important. This idea is further supported by our observation of a largely density independent behavior of astrocytes, that were jammed at all analyzed cell densities. However, during the scratch-assay, astrocytes displayed the expected initial fluidization [53,54], but upon contact of the opposing fronts, the velocity field at the meeting points oriented roughly perpendicular to the opposing fronts, or movement ceased and cells from both fronts did not mix. Such a behavior was reported previously in MCF-10A cells and associated with the induction of jamming [55]. Taken together, reaching confluence might be sufficient to induce jamming and migratory arrest in astrocytes. Another key difference between astrocytes and GBM cells was the ability of both GBM cell lines to proliferate. Previous studies found a positive association between proliferation and the ability to migrate [56,57,58,59,60,61]. Proliferation events were supposed to induce local stress fluctuations, and even fluidization of cell layers [56,57,58,59,60,61]. Consequently, the proliferative ability of the here investigated GBM cells might potentially be an additional source of the migratory, unjammed state. Nonetheless, the detailed influence of proliferation in the analyzed system is not clear and needs further investigations. Especially, given the cell density independent behavior of U138 cells. Furthermore, differences in energy metabolism between astrocytes and GBM cells have to be taken into account, as GBM show higher rate of glycolysis [62,63]. This is especially noteworthy, because unjamming was previously demonstrated to be associated with a shift towards glycolysis, being another factor potentially explaining the differences between both cell types [64,65]. 

As cell density might not be the sole control parameter for the jamming transition in GBM cells, we next turned to the predictions made by other models [66]. These models propose that cell–cell-adhesion, tension and propulsion forces determine collective behavior [26,27,66]. Of note: The different effects of both inhibitors on single cell motility and speed of the confluent monolayers are a first and strong hint that not only single cell properties, but also cell–cell interactions, such as adhesion, are affected by the used inhibitors. To test the predictions of the aforementioned models, the force generation was disturbed by the inhibition of myosin II and ROCK. Notably, force generation is coupled with adhesion and tension [51,52], as is reproduced here as well. As expected, all but one treatment led to a reduced migration speed in the cell monolayer [67,68,69] and lowered reorganization in all cell systems. Interestingly, in LN229 cells, the lifetime of fast moving packs almost approached those of astrocytes, hinting towards the onset of jamming. This idea was further supported by the shape change leading to a drop of the shape factor of LN229 cells reaching the critical value predicted for jamming [26,27]. U138 cells reacted qualitatively similar, but in all analyzed metrics, remained far from jamming. In parallel to the migratory arrest, we found cell–cell adhesion to be stronger affected in LN229 cells than tension, while this was not the case for U138 cells that stayed motile. Such behavior is in support of the predictions made by the vertex model and other experiments [19,26,27,28]. Yet, it has to be considered that cell–surface interactions most likely were reduced by the used interventions too. Consequently, their role needs to be elucidated, as they are neither included in the vertex model nor explicitly measured here. 

Furthermore, a difference in interpretation of the role of stress fibers compared to a study of Saraswathibhatla and Notbohm should be considered [25]. In their study performed in Madin-Darby canine Kidney (MDCK) cells, they argued that more stress fibers led to more traction and thus to more reorganization and an increased shape factor, associating the stress fibers directly with propulsion forces and thus being the main inducer of (un-)jamming. Contrasting this idea, a different study found stress fibers in U2OS osteosarcoma cells to carry no more than 20% of the overall forces [45], yet these values may differ depending on the cell type. Nevertheless, here we considered stress fibers as part of the tension machinery, thus favoring less elongated cell shapes and jamming. This concept is supported by observations made using laser ablation of single stress fibers, leading to almost immediate (<2 min) expansion of single cells [46,70], indicating that stress fibers also function as a restraining element. Furthermore, stress fiber contractility was suggested even being inhibitory to migration and consequently, inhibition of stress fiber contractility might even increase cell motility [71,72,73]. Additionally, our measurements implied that the number of stress fibers per unit area between both GBM cell lines was comparable for the respective treatments and thus traction created. Consequently, in our experimental setting, stress fiber traction was considered to be unlikely a part of the propulsive forces. Nonetheless, we can neither fully rule out this possibility, nor the idea that the restraining forces created by the stress fibers help to build up larger propulsive forces to overcome the energy barrier imposed on cells by their neighbors. Independently on the exact role of stress fibers, the presented data shows the induction of jamming in LN229 cells treated with myosin II or ROCK inhibitor, accompanied by a changed ratio of adhesion to tension in favor of tension. 

So far, all discussed friction parameters were equilibrium parameters defined via time scales of hours to days. Yet, at high cell densities, a migrating cell needs to elongate and thereby detach old adhesions and form new ones, happening on a time scale of seconds to minutes. The performed adhesion measurements using the AFM were conducted on a time scale of one minute. At these time scales, adhesive forces were significantly less impacted by the used treatments (relative reductions: LN229: ≤20%; U138: ≤55%;), albeit the effect of the drugs was likely underestimated as the calculations did not account for the softening of the cell monolayer after treatment and thus the larger contact area. Notably, cell–cell adhesion on this time scale might be less affected than tension, implicating traction forces to be the main driver of (un-)jamming [25,66]. 

Another aspect reminiscent of a jamming transition is a solidification of the jammed matter that was not observed here. Treatment with blebbistatin or Y-27632 inhibited force generation, caused by the dissociation of most stress fibers and disturbance of the actin cortex [37,74]. The observed lower elastic modulus is suspected to be caused by the induced structural changes of stress fibers, the actin cortex and other structures significantly contributing to the measured elastic modulus [75,76] and a loss of intracellular tension reducing strain hardening of actin and/or intermediate filament [77,78,79,80]. 

Taken together, our data demonstrated that astrocytes reside in a jammed state, while GBM cells are unjammed. A disturbance of the equilibrium of tension and cell–cell-adhesion can lead to jamming in GBM cells, effectively hindering layer migration and potentially infiltration into adjacent tissue, but also cell density can be a driver of jamming in GBM cells. These findings shed new light onto the collective migration of GBM and may open up new roots for the development of anti-migratory agents in glioblastoma. 

## Figures and Tables

**Figure 1 cells-12-00029-f001:**
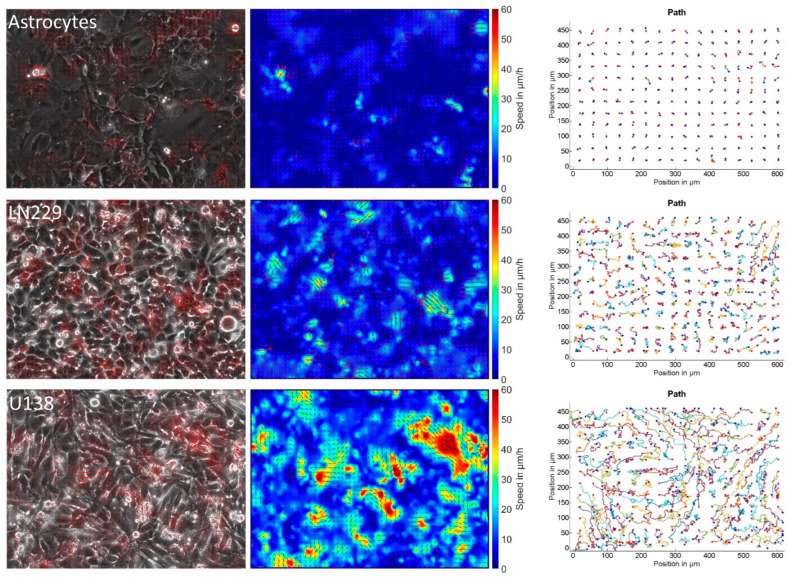
Migration of astrocytes and glioblastoma cells. Each row shows a typical phase contrast image, overlaid with the vectors of the velocity fields (**left**), the magnitude of the velocity and its direction (**center**) and typical cellular movement over the time course of 20 h (**right**). Red dots denote the start and blue dots the end position.

**Figure 2 cells-12-00029-f002:**
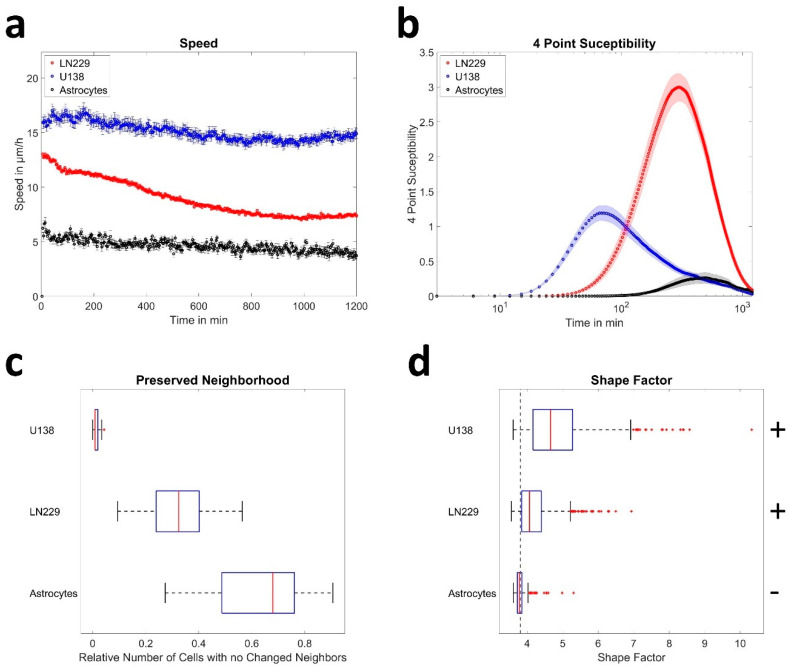
Collective migration properties of astrocytes and glioblastoma cells. (**a**) Graph of the mean speed per field of view over time, evaluated for 81, 33 or 18 fields of views for LN229, U138 or astrocytes, respectively. (**b**) 4-point susceptibility as obtained from the velocity fields over the whole measurement time. Peak positions of the 4-point susceptibility represent the average life time of collectively moving packs of cells. Sample sizes were identical as reported in (**a**). (**c**) Analysis of layer reorganization in terms of cells without making new neighbors. Sample sizes were identical as reported in (**a**). (**d**) Differences in shape factor between the cell types measured during live-cell imaging. The dotted line shows the critical value of 3.81 predicted by the vertex model. Below cells are considered to be jammed. “+” and “−” depict if shape factors are significantly larger or smaller than the critical value, with *p* < 0.0001 for all cell types, as tested with sign test. For shape factor calculation 2645, 772 or 269 LN229, U138 cells or astrocytes were measured from the fields of view. (**b**,**c**) Error bars and shaded areas depict the standard error of the mean. (**c**,**d**) Box plots show the median (red line), 25 and 75 percentile (box), non-outlier range (whiskers) and outliers (red dots).

**Figure 3 cells-12-00029-f003:**
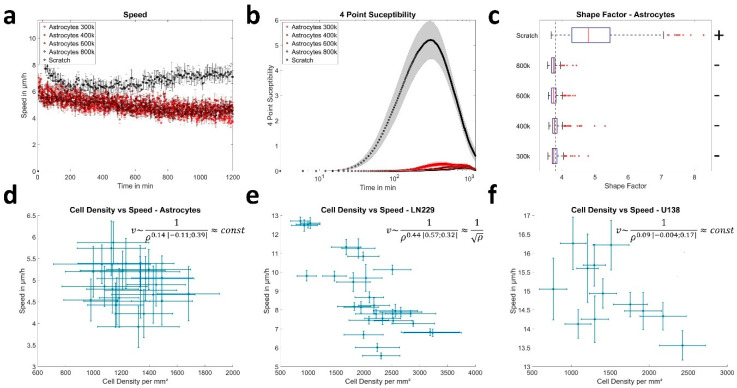
Cell density dependence of collective migration. (**a**) Measured mean speeds of astrocytes per field of view when seeded in different densities of 300k, 400k, 600k, 800k cells or subjected to a scratch, measured for 18 or 20 (scratch) fields of view. (**b**) Illustration of the 4-point susceptibility over the whole measurement time. Peak positions of the 4-point susceptibility represent the average life time of collectively moving packs of cells. Sample sizes are identical to (**a**). (**c**) Illustrates the difference in shape factor for different seeding densities of astrocytes and during the wound healing assay. The dotted line shows the critical value of 3.81 predicted by the vertex model. Below cells are considered to be jammed. “+” and “−“ depicts if shape factors are significantly larger or smaller than the critical value, with *p* < 0.0001 for all instances, as tested by sign test for 256, 269, 234, 183 or 386 astrocytes for seeding densities of 300k, 400k, 600k, 800k cells or the scratch. (**d**–**f**) Plot of the cell density over the speed for astrocytes, LN229 and U138 cells. Formulas show the scaling behavior of the speed v with cell density δ. Values in brackets denote 95% confidence intervals. (**a**,**b**,**d**–**f**) Error bars and shaded areas depict the standard error of the mean. (**c**) Box plots show the median (red line), 25 and 75 percentile (box), non-outlier range (whiskers) and outliers (red dots).

**Figure 4 cells-12-00029-f004:**
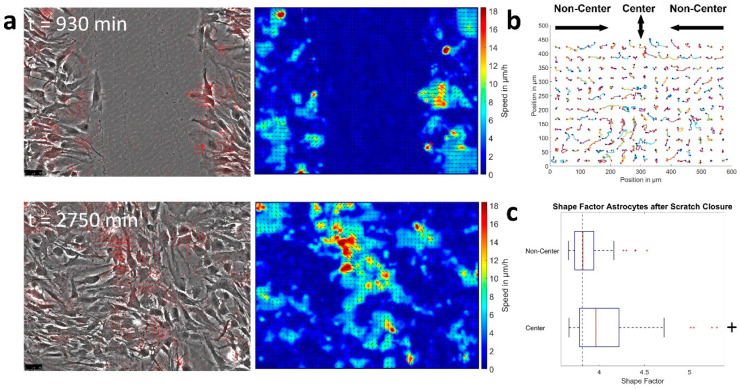
Behavior of astrocytes after wound closure. (**a**) Shows phase contrast images of astrocytes subjected to a scratch at different time points after the injury, overlaid with the velocity vectors (left). The right part shows the local speed and its direction. (**b**) Depiction of the movement paths taken by the cells of (**a**) after scratch closure. Red dots denote the start and blue dots the end position. (**c**) Illustration of the difference in shape factor for two distinct regions (see (**b**)) of the wound healing assay after scratch closure. The dotted line shows the critical value of 3.81 predicted by the vertex model. Below cells are considered to be jammed. “+” depicts if shape factors are significantly larger or smaller than the critical value with *p* = 0.049, as tested by sign test for 61 or 85 cells at the center or in the non-center region. Box plots show the median (red line), 25 and 75 percentile (box), non-outlier range (whiskers) and outliers (red dots).

**Figure 5 cells-12-00029-f005:**
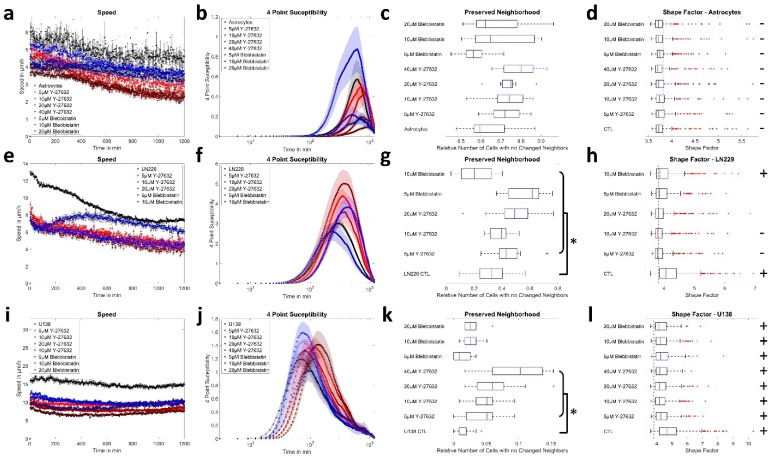
Effect of myosin II and ROCK inhibition on collective migration. (**a**,**e**,**i**) Mean speed per field of view for astrocytes, LN229 and U138 cells treated with ROCK inhibitor Y-27632 or myosin II inhibitor blebbistatin. For astrocytes, 15 fields of view were measured for all conditions. For LN229 and U138 cells 81 or 33 fields of view were assessed in control conditions and 15 otherwise. (**b**,**f**,**j**) 4-point susceptibility over the whole measurement time for astrocytes, LN229 and U138. Peak positions of the 4-point susceptibility represent the average life time of collectively moving packs of cells. Sample sizes are the same as in (**a**,**e**,**i**), respectively. (**c**,**g**,**k**) cell layer reorganization in terms of changed neighborhood. Sample sizes are the same as in (**a**,**e**,**i**), respectively. Here *p* < 0.001 for all instances of U138 and LN229 cells except for LN229 treated with 10 µM Y-27632 (*p* < 0.05). (**d**,**h**,**l**) illustrate the difference in shape factor for astrocytes, LN229 and U138 cells. The dotted line shows the critical value of 3.81 predicted by the vertex model. Below cells are considered to be jammed. “+” and “−” depict if shape factors are significantly larger or smaller than the critical value, as assessed by sign test. Here *p* < 0.0001 holds for all instances of astrocytes, U138 cells and LN229 cells in control conditions or when treated with 10 µM blebbistatin. *p* < 0.01 for LN229 cells treated with 5 µM Y-27632 and *p* < 0.05 for LN229 treated with 10 µM Y-27632. For astrocytes 365, 284, 261, 279, 296, 390, 413 or 396 cells were measured for control conditions, Y-27632 (5, 10, 20, 40 µM) or blebbistatin (5, 10, 20 µM) treatment. For LN229 cells 2645, 298, 301, 454, 302 or 460 cells were measured for control conditions, Y-27632 (5, 10, 20 µM) or blebbistatin (5, 10 µM) treatment, respectively. For U138 cells 772, 448, 448, 450, 150, 300, 302 or 150 cells were measured for control conditions, Y-27632 (5, 10, 20, 40 µM) or blebbistatin (5, 10, 20 µM) treatment, respectively. (**a**,**b**,**e**,**f**,**i**,**j**) error bars and shaded areas depict the standard error of the mean. (**c**,**d**,**g**,**h**,**k**,**l**) box plots show the median (red line), 25 and 75 percentile (box), non-outlier range (whiskers) and outliers (red dots). Stars (*) depict statistically significant results.

**Figure 6 cells-12-00029-f006:**
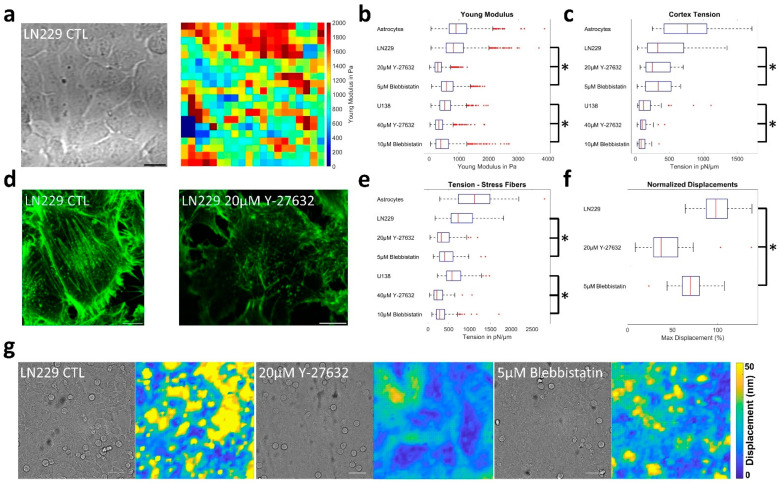
Biomechanical properties of astrocytes and glioblastoma cells. (**a**) Illustration of a typical 140 × 140 µm^2^ dense layer of LN229 cells used for measuring elastic moduli (left) and the respective map of the elastic moduli (right). Scale bar represents 20 µm. (**b**) Measurements of the Young moduli for astrocytes and both GBM cell lines, when treated with the ROCK inhibitor Y-27632 or myosin II inhibitor blebbistatin. *p* < 0.0001 for all significant results, as assessed by one way ANOVA with Tukey post hoc test. For astrocytes 1187 sample curves were obtained. For LN229 1568, 1566 or 1585 and for U138 cells, 1161, 1164 or 1148 measurement curves were evaluated for control conditions, Y-27632 or blebbistatin treatment, respectively. (**c**) Cortical tension of astrocytes, LN229 and U138 cells, when subjected to ROCK or myosin II inhibition. *p* < 0.05 for all LN229 measurements and U138 control conditions vs. 10 µM blebbistatin. *p* < 0.01 for U138 vs. 40 µM Y-27632. One way ANOVA with Tukey post hoc test was used. For astrocytes, 45 cells were measured. For LN229, 55, 55 or 53 and for U138, 55, 53 or 54 cells were measured for control conditions or when treated with Y-27632 or blebbistatin, respectively. (**d**) A typical actin staining of LN229 cells in control conditions (left) and when treated with 20 µM of Y-27632 is shown. Scale bars depict 10 µm. (**e**) Estimation of the tension generated by stress fibers by astrocytes and GBM cell lines. 75 cells in 15 fields of view were analyzed in each group. *p* < 0.0001 for all significantly different treatments, as assessed by one way ANOVA with Tukey post hoc test. (**f**) Normalized bead displacements obtained from the TFM measurements for LN229 cells. *p* < 0.0001 for all significantly different treatments. (**g**) Sample images and the respective displacement maps for (un-)treated LN229 cells. Scale bars depict 50 µm. (**b**,**c**,**e**,**f**) Box plots show the median (red line), 25 and 75 percentile (box), non-outlier range (whiskers) and outliers (red dots). Stars (*) depict statistically significant results.

**Figure 7 cells-12-00029-f007:**
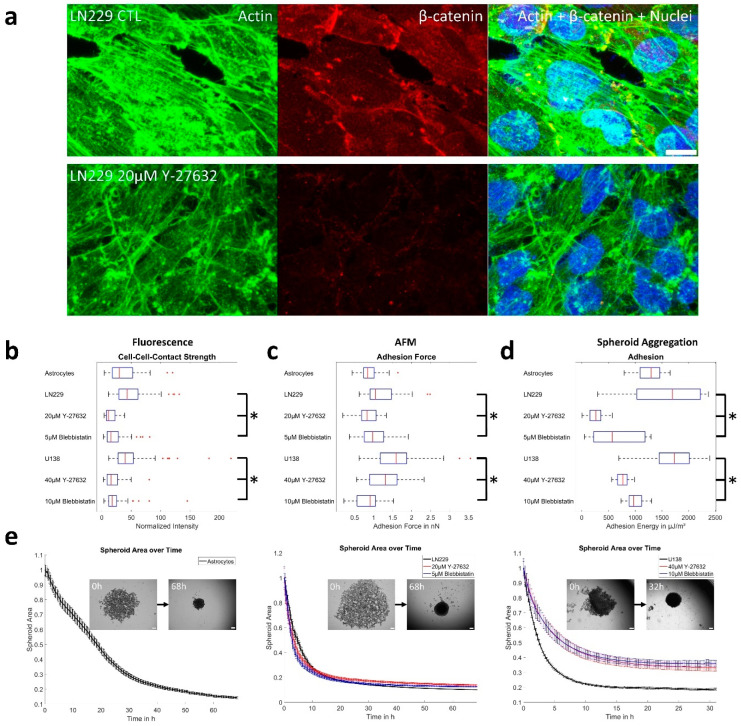
(**a**) Representative examples of actin and β-catenin staining for LN229 cells. Scale bar depicts 10 µm. (**b**) Cell–cell adhesion strength as estimated by the fluorescence images. For all statistically significant differences: *p* < 0.0001. For astrocytes 73 cell–cell connections were measured. For LN229 70, 49 or 72 and for U138 75, 74 or 67 cell–cell junctions were analyzed for control conditions or when treated with Y-27632 or blebbistatin, respectively. (**c**) Cell–cell adhesion forces as measured using AFM. For LN229 treated with Y-27632 and U138 treated with blebbistatin *p* < 0.01, *p* < 0.05 otherwise. The sample size for astrocytes was 30, for LN229 39, 35, 34 and for U138 32, 27, 26 for control conditions or after treatment with Y-27632 or blebbistatin, respectively. (**d**) Values calculated for the cell–cell-adhesion energies obtained from the spheroid aggregation. *p* < 0.0001 for all significantly different treatments, as assessed by one way ANOVA with Tukey post hoc test. For astrocytes 46 spheroids were assessed. For LN229 31, 23 or 17 and for U138 cells 33, 39 or 27 spheroids were measured for control conditions or Y-27632 or blebbistatin treatment, respectively. (**e**) Evolution of the spheroid area over time for astrocytes and both GBM cell lines for all used conditions. Inlets show typical spheroids at the beginning and end of the measurement. Sample size is identical to d. Scale bars depict 300 µm. (**b**–**d**) Box plots show the median (red line), 25 and 75 percentile (box), non-outlier range (whiskers) and outliers (red dots). (**e**) Error bars depict the standard error of the mean. Stars (*) depict statistically significant results.

**Table 1 cells-12-00029-t001:** Ratio of median adhesion and tension to control conditions.

Treatment	Tension to CTL	Adhesion to CTL ^#^
LN229 + Y-27632	38%	26%/16%
LN229 + Blebbistatin	71%	35%/33%
U138 + Y-27632	38% *	40%/44%
U138 + Blebbistatin	49% *	44%/56%

* indirect measurements. # Results of fluorescence quantification and spheroid aggregation.

## Data Availability

The published article includes all datasets generated or analyzed during this study. The code supporting the current study has not been deposited in a public repository but is available from the lead author on request.

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
