# Peer review of "Jamming Transitions in Astrocytes and Glioblastoma Are Induced by Cell Density and Tension"

_cells, 2022, doi:10.3390/cells12010029_

Round 1

Reviewer 1 Report

The manuscript by Hohmann et al. aimed to dissect out the collective behavior of primary astrocytes and GBM cells. They found astrocytes were in a jammed state while GBM cells kept unjammed with marked reorganization under in vitro cell culture conditions. Next, they investigated the roles of cell density on cell jamming in a cell line dependent manner. Lastly, they discovered that in a GBM cell line, a switch from unjammed to jammed status can be controlled by the equilibrium of tension and cell-cell adhesion. The authors showed comprehensive data with decent quantification, provided a very interesting phenomenon and  the related mechanism, I suggest this article can be accepted in principle if the authors can address the following points.

1)      In 3.2 section, the authors investigated the effects of cell density on jamming with GBM cell lines from low density to high density due to cell proliferation. However, it is unknown whether cell-dividing can interfere the measurement or not. Could the authors perform the assays with different density of GBM cells treated with cell-dividing blockers? For the astrocyte density, the range is a bit of small, could the authors increase the density range like 20k, 100k, 300k, 800k?

2)      On line 366, please indicate the panel of Figure 3 instead of “Figure 3”.

Author Response

Dear reviewer,

thank you for your time an comments. Please find the detailed response attached.

Reviewer 2 Report

The study describes the transition mechanics/biophysics of astrocytes as they become migratory or "unjammed" to spread throughout the brain in glioblastoma. This is a novel biophysical investigation on the origins of cancer formation. Minor comments:

1. The title could be enhanced by including something about expansion or migration of cells 

2.  The abstract could be expanded to include a bit more about the actual techniques used, and a bit more about the magnitude or data findings. At first it seemed like a review manuscript. 

3. Line 66: It might be direct to also include that jamming is nonmigratory cells while unjammed is strong propensity to migrate; gives better goal of what is being measured. 

4.  In the discussion, how would the differences in primary cila, and mitochondrial support of the actual movement support the differences seen?

Author Response

(The authors gave the same response as above.)

Reviewer 3 Report

Hohmann et. al. provides a thorough in vitro study that proposes astrocytes and GBM cells to exist in either a jammed or unjammed state. Such a transition is exactly what normal cells have been proposed to undergo during EMT, which has implications in the cancer field. To validate this notion the authors utilize several biophysics-based approaches to characterize differences in the physical state of a primary astrocyte line and 2 GBM cell lines. While a significant amount of data is provided to support this idea and this reviewer did view this article in a positive manner, there are questions with some of the techniques and data presented here that need to be addressed. Questions and comments are as follows;

supplementary figures and supplementary video was not uploaded. Until this is done and this data is seen it is difficult to completely assess the entire paper as this data should fill gaps in the paper. Please make sure this is added as there are multiple references to supplementary videos and figures not present.

Please check for typos through the paper. For example, “pg 1 line 31 - month should be months”, pg 4 line 165 - should title be “3D migration?”

Please list passage number used for astrocyte testing

How are readers confident that what they are measuring is actually changes in cell-cell adhesions trench vs. stiffness changes

pg 4 lines 152 - 162 -  This section states that a tip-less cantilever was pressed onto a cell in suspension (that’s non adherent) at 1 nN for 30s, same cell was then allowed to attach to another cell for 30 min. From this the elastic modulus and  cell-cell adhesion force was measured by pressing cantilever on cell for 60s on multiple spots of a dense cell layer at force of 0.5 nN. There is inconsistency in force being used for single cell (1nN) and force being used to once cell is attached to other cells (0.5 nN). A few questions regarding this 1) why are different forces used? 2) was the same cell used in this instance, 3) If this reviewer is to understand this correctly afm was done on cells in suspension, which seems extremely technically difficult. With that being said how are the authors confident that what they are measuring are actually cell-cell adhesion forces?

pg4 lines 166 - please explain what the liquid overlay method is.

p4 178 - what is a equilibrium projected spheroid area as well as an equilibrium brightness. Authors need to describe what conditions they are defining when an equilibrium is reached. What are the guidelines for this, for example?

p5 line 186 - d is initial cell distance of what? Are the authors tracking the same cell for the entire 68h experiments?

p5 line 189 - authors states compact “describes the geometry of the 3D aggregate”. This is very abstract and does not give this reviewer or any reader in general what this means. Please provide a bit more detail on what this is. Is this a dimensionless number or are there units? What does a aggregate being more or less compact mean for aggregate geometry?

p5 line 198 - authors need to explain how they measured aggregate area and height. In general AFM can be used to make topographical measurements, but using AFM to make such measurements of a 3-D spheroid seem to be technically challenging if not impossible. More detail needs to be given on how this was done. This reviewer assumes confocal was used, but its not clear.

In what context or experimental conditions was stress fiber tension measured? if actin was stained with phaolloidn the cells were fixed at that point so how can the authors be confident that differences they observed are not a function of the cells being fixed, if this was the case?

What the authors have listed in the “Traction Force Microscopy” section  in “Methods” are not tractions. This simply describes measuring bead displacements. While the tractions will in general follow the behavior of the bead displacements the heat maps of bead displacements themselves are not tractions. This section needs to be corrected as  what the authors describe here are not tractions. 

what is SIFT algorithm? there’s no reference that can be referred to or explanation of this. Authors shouldn’t assume readers will know what this is.

astrocytes and U138 moved in packs of 2 -3 cells, but LN229 moved in packs of 10 do authors have explanation for this?

p9 line 362 - authors mention no significant proliferation was observed for astrocytes after 20h but there is no evidence of this provided or even a mention of how proliferation was measured. Was this proliferation observation quantitative or qualitative? This information needs to be added.

 p9 line 363 - what different densities of astrocytes were seeded? saying that they were different is insufficient. Were the densities orders of magnitude different? Without any numbers its difficult to have any reference of comparison for densities.

A scratch  was created to generate initial density of zero for astrocytes, but what about GBM cells? Why not do the same scratch zero control for GBM cells as well?  Without the same type assays to compare for each cell type how do the authors expect readers to make any reasonable comparisions?

GBM cells showed higher proliferation, but again was this something measured quantitatively or a qualitative observation? Whatever the case may be the authors need to be more specific about what was done.

figure 3 - text in all plots are very small and its difficult to see please enlarge to make more readable. Also for this figure what does astrocytes “300k” “400k” etc. mean? is this number of cells or cell density? Also, its strange that this figure A - C are not done for the other GBM cell lines as well. Why show cell speed for just the astrocytes with no comparison for the GBM cell lines?

For bleb and Y27632 experiments its stated that drugs were added 1 hour proper to experiments, but was a washout done or was imaging done in presence of these drugs?

figure 5 please increase font size in figures it’s very difficult to see anything.

Looking at  figures for preserved neighborhood and shape factor there is no consistency in bleb and Y27632 concentrations used. For example,  5C (Astrocytes) uses a max bleb concentration of 20, 10, and 5 while 5G (LN229) uses  bleb concentrations of 10, and 5, and 5K (U138) uses  bleb concentrations of 20 and 10. The same can be said for the ROCK inhibitors as well. With so many inconsistent concentrations used its difficult to make sense of the point authors are trying to get across as there is only one concentration (10) that is comparable.

figure 6A -B - this elastic moduli map looks like LN229 cells are adherent, but this it was mentioned that AFM was done with cells in suspension for cell-cell adhesion experiments? If this is the case why were different measurements done with some cells adhered to substrate vs. other being non-adhered?

The authors state that bead displacements are lowered and thus tractions are lowered, while this is true the authors assume that this is common knowledge to anyone not an expert in this technique. There are not actual traction values provided and there is not even a description of what tractions are and how they relate to substrate displacements. What the authors did is  PIV to tract beads displacements and the authors should just leave it at that. Tractions are in units of Pa, there are no actual traction force measurements provided anywhere in this paper. Unless the authors have provided traction measurements this should be removed.

Authors state U138 cells did not attach to laminin-coated gels, but what about astrocytes? Why were bead displacements not done for astrocytes as well? This could have easily been done for astrocytes to at least provide a comparison between displacements generated by “jammed” and “unjammed” cells. What exactly are readers supposed to appreciate from bead displacements of one cell type? Without a comparison to other cell types this reviewer has difficulty seeing the value the bead displacement color plots add taken into account there are already plots showing this for all cell types as well (fig 6f).

for figure 6  why was concentration of  only 20 for Y27632 and 5 for ble used for LN229 cells?  These both represent upper and lower limits of the drug stimulations used, respectively. Why not do the two upper limits or two lower limits?  

in some of the phase images many of the cells look rounded up. How are the authors confident cells are not dead vs just changing their shape? Did the authors do any type of live/dead assay? The authors need to provide some type of confidence that cells were still alive and active.

What solvent was bleb and Y27632 diluted in? DMSO, for example? This reviewer ask as depending on concentration of solvent used what authors could be seeing is an impact of solvent used to dilute drug vs the drug itself. The authors should provide what the drugs were diluted in and final concentrations to confirm that solvent has no impact on results.

figure 7B why are we provided fluorescent based “Cell-cell contact strength” data for all 3 cell types at multiple concentrations, but actual fluorescent images for only LN229? Please provide representative images for the other 2 cell types along with comparable concentrations. Whether this is in the main figures or supplementary this should be provided.

Author Response

Dear reviewer,

thank you for your time and comments. Please find the detailed response attached.
